# Carbidopa Alters Tryptophan Metabolism in Breast Cancer and Melanoma Cells Leading to the Formation of Indole-3-Acetonitrile, a Pro-Proliferative Metabolite

**DOI:** 10.3390/biom9090409

**Published:** 2019-08-24

**Authors:** Diana Duarte, Filipa Amaro, Isabel Silva, Dany Silva, Paula Fresco, José C. Oliveira, Henrique Reguengo, Jorge Gonçalves, Nuno Vale

**Affiliations:** 1Laboratory of Pharmacology, Department of Drug Sciences, Faculty of Pharmacy, University of Porto, Rua de Jorge Viterbo Ferreira, 228, 4050-313 Porto, Portugal; 2Institute of Molecular Pathology and Immunology of the University of Porto (IPATIMUP), Rua Júlio Amaral de Carvalho, 45, 4200-135 Porto, Portugal; 3Instituto de Investigação e Inovação em Saúde (i3S), University of Porto, Rua Alfredo Allen, 208, 4200-135 Porto, Portugal; 4Clinical Chemistry, Department of Laboratory Pathology, Centro Hospitalar Universitário do Porto (CHUP), Largo Prof. Abel Salazar, 4099-313 Porto, Portugal; 5Unit for Multidisciplinary Research in Biomedicine (UMIB), Abel Salazar Biomedical Sciences Institute (ICBAS), University of Porto, Rua de Jorge Viterbo Ferreira, 228, 4050-313 Porto, Portugal; 6Department of Molecular Pahology and Immunology, Abel Salazar Biomedical Sciences Institute (ICBAS), University of Porto, Rua de Jorge Viterbo Ferreira, 228, 4050-313 Porto, Portugal

**Keywords:** indole-3-acetonitrile, tryptophan, carbidopa, MCF-7 cell line, A375 cell line

## Abstract

Carbidopa is used for the treatment of Parkinson’s disease (PD) as an inhibitor of DOPA decarboxylase, and PD patients taking carbidopa have a lower incidence of various tumors, except for breast cancer and melanoma. Recently, it was shown that carbidopa inhibits tryptophan-2,3-dioxygenase (TDO) and kynureninase enzymes. In the present study, the effect of carbidopa on the viability and metabolic profile of breast cancer MCF-7 and melanoma A375 cells was investigated. Carbidopa was not effective in inhibiting MCF-7 and A375 proliferation. Liquid chromatography and mass spectrometry revealed a new compound, identified as indole-3-acetonitrile (IAN), which promoted a concentration-dependent increase in the viability of both cell lines. The results suggest that treatment with carbidopa may alter tryptophan (Trp) metabolism in breast cancer and melanoma leading to the formation of a pro-proliferative Trp metabolite, which may contribute to its failure in reducing breast cancers and melanoma incidence in PD patients taking carbidopa.

## 1. Introduction

Tryptophan (Trp) metabolism is frequently altered in cancers, since Trp is catabolized in the microenvironment of tumours and in inflammatory and other immune-privileged sites. In addition to its involvement in the production of serotonin (5-HT), 95% of dietary Trp is metabolized along the kynurenine pathway (KP), which is recognized as an important link between inflammation and neoplastic progression in several types of cancer [1]. The conversion of Trp to kynurenine (Kyn) is catalysed by indoleamine-2,3-dioxygenase 2 (IDO1), a well-established target for drug discovery in cancer immunotherapy [2]. Two other enzymes metabolize Trp along the KP: indoleamine-2,3-dioxygenase 2 (IDO2) and tryptophan-2,3-dioxygenase (TDO) [3]. All three enzymes have been shown to be expressed in a variety of cancers, and recent studies have pointed out the relevance of TDO in some cancers (namely, malignant glioma, melanoma, and bladder cancer), where this enzyme is constitutively expressed [4]. 

Carbidopa is primarily used for the treatment of Parkinson’s disease (PD) to spare L-DOPA from peripheral inactivation. Recently, it has been proposed for the treatment of pancreatic cancer as a repurposed drug [5]. Carbidopa is known as an inhibitor of DOPA decarboxylase. It was further shown that carbidopa alters Trp metabolism by inhibiting TDO and kynureninase. Therefore, carbidopa may have direct effects on treated PD by modifying tryptophan metabolism [6]. Treated PD patients have a low incidence rate of various types of cancer [7]. Breast cancer and melanoma are exceptions and present an uncharacteristically high rate of co-occurrence with PD treatment [8]. The question arises of whether carbidopa is disturbing some balance in Trp metabolism, favouring the formation of pro-proliferative Trp metabolites promoting breast cancer and melanoma, thus causing a different incidence profile of these tumours in treated PD patients. 

In this study, carbidopa effect was tested in breast cancer (MCF-7) and melanoma (A375) cell lines. The extracellular medium of cell cultures, exposed and not exposed to different concentrations of carbidopa within the therapeutic range for PD [9], was analysed by LC-MS. A novel Trp metabolite was identified, indole-3-acetonitrile (IAN), formed from an alternative metabolic pathway. IAN is synthesised by bacteria and plants under attack as a defence and survival messenger [10]. The present study describes, for the first time, the synthesis of IAN in human cancer cell lines. IAN was shown to increase the viability of these cell lines, which may indicate a role of this Trp metabolite in promoting breast cancer and melanoma in PD patients treated with drugs containing carbidopa.

## 2. Materials and Methods 

### 2.1. Cell Viability Assays

Human A375 melanoma cells and MCF-7 breast cancer cells (ATCC, American Type Culture Collection, Manassas, VA, USA) were seeded in 96-well plates (200 μL per well), with an initial cell density of 3.9 × 10^4^ cells/mL and 3.0 × 10^4^ cells/mL, respectively. The cells were allowed to attach for 24 h and were then submitted to a 3 h serum starvation period. The cells were next either left untreated (exposed only to vehicle, i.e., to DMSO at maximum final concentration of 0.1% v/v) or treated with carbidopa (15 µM) or IAN (1, 15 or 30 µM) for 24 h. During the experimental period, the cells were maintained at 37 °C with 5% CO_2_. Cell viability was evaluated using the MTT protocol. At the end of the incubations, the cell medium was removed, and 100 µL/well of MTT solution (0.5 mg/mL in PBS) was added; the cells were incubated for 3 h, protected from light. After this period, the MTT solution was removed, and DMSO (100 µL/well) was added to solubilize the formazan crystals. Absorbance was measured at 570 nm in an automated microplate reader (Sinergy HT, Biotek Instruments Inc, Winooski, VT, USA). The results are expressed as percentage of the respective control (vehicle). All conditions were performed in sextuplicate.

### 2.2. LC-MS Analysis

For the measurement of metabolites release from the MCF-7 and A375 cells, the samples were separated on a HPLC Accela (Thermo Fischer Scientific, Bremen, Germany) using an ACE Equivalence C18 column (VWR/Avantar, Alfragide, Portugal), 5 µm particle size and dimensions of 3.0 mm ID × 75 mm. The samples were eluted over a gradient of 100% solvent A (CH_3_COOBH_4_, 10 mM) to reach 100% Solvent B (acetonitrile with HCOOH 0.1%) during 8 min at a flow rate of 0.4 mL/min. Analyses were done on an LTQ OrbitrapTM XL hybrid mass spectrometer (Thermo Fischer Scientific, Bremen, Germany) controlled by LTQ Tune Plus 2.5.5 and Xcalibur 2.1.0. The capillary voltage of the electrospray ionization source (ESI) was set to 3.1 kV. The capillary temperature was 275 °C. The sheath gas and auxiliary gas flow rate were 40 and 10 (arbitrary units, as provided by the software settings). The capillary voltage was 32 V, and the tube lens voltage was 55 V. An MS data handling software (Xcalibur QualBrowser software, Thermo Fischer Scientific, Waltham, MA USA) was used to search for metabolites by their *m/z* value.

### 2.3. HPLC Analysis

Serotonin was determined by an HPLC-ECD method using the 3030 Reagent kit for HPLC analysis of serotonin in serum/plasma/whole blood from Chromsystems GmbH, Munich, Germany, and the 6000 Reagent kit for HPLC analysis of serotonin in urine, respectively, as described by the manufacturer. The HPLC system was a Waters Alliance^®^ 2695 pump (Waters Corporation, Milford, MA 01757, USA) with a Rheodyne loop injector and a Decade (Antec Scientific, 2382 NV Zoeterwoude, Netherlands) electrochemical detector with a glassy carbon electrode set to a potential of + 0.75 V with reference to a saturated KCl-filled Ag/AgCl reference electrode, and the sensitivity of the detector was set to 5 nA. Reversed-phase chromatography was carried out with the 6100 and 3130 HPLC columns recommended for these kits (Chromsystems GmbH, Gräfelfing/Munich, Germany). The mobile phase used was the one suggested by the manufacturer. The samples were injected onto the column via a Perkin Elmer autosampler series 200 (PerkinElmer, Villepinte, France) with a cooling tray set at 4 °C. The HPLC measurements were performed at 37 °C with 1.000 mL/min flow rate. The current produced was monitored by Empower Pro (Waters Corporation, Milford, MA 01757, USA) software. A calibration curve in the concentration range of 1.0 to 30.0 nM for serotonin was constructed by plotting the peak area of standard samples. The amounts of serotonin in each cultural medium were calculated from the calibration curve of serotonin standards. 

### 2.4. Statistical Analysis

The results are presented as mean ± SEM for n experiments performed. Statistical comparisons between groups, at the same time point, were performed with One-Way ANOVA, after Shapiro–Wilk test normality evaluation. Significance was accepted at *p* values < 0.05. The Student–Newman–Keuls post-hoc test was used once a significant *p* was achieved.

## 3. Results

### 3.1. Involvement of Trp Metabolites in Cancer

The metabolism of Trp is one of the routes that is altered in several types of cancer. In order to evaluate the metabolic ability of A375 and MCF-7 cells and to identify the production of Trp metabolites in tumour models, metabolites produced by the Kyn pathway were investigated by mass analysis (LC-MS), namely, *N*-formylkynurenine, Kyn, kynurenic acid, anthranilic acid, quinolinic acid, 3-hydroxykynurenine and xanthurenic acid (Figure 1 and Figure 2, respectively). 

Serotonin production was investigated by HPLC-ECD quantification. Both MCF-7 and A375 were able to metabolize Trp to serotonin (Figure 3A,B, respectively). However, when considering Trp metabolites formed via Kyn pathway, the medium from A375 cells presented less metabolites (only kynurenic acid and anthranilic acid) compared with that of MCF-7 cells, in which several intermediates were detected by LC-MS (Figure 4).

### 3.2. Exposure of A375 and MCF-7 to Carbidopa 

The viability of MCF-7 and A375 cells was not reduced by exposure to 15 µM carbidopa (Figure 5A,B, respectively). In MCF-7 cells, even a significant increase in cell viability was observed (Figure 5A).

Metabolic screening, by LC-MS analysis of the incubation medium of MCF-7 and A375 cells exposed and not exposed to carbidopa (15 µM), revealed the presence of a new metabolite produced by A375 and MCF-7 cells exposed to carbidopa: IAN, with *m/z* 157.04. Its relative abundance was higher in A375 cells after 0.5 h incubation with carbidopa (Figure 6A) but increased over time in MCF-7 cells (Figure 6B): after a 24 h incubation, the relative abundance of IAN was higher in MCF-7 than in A375 cells.

### 3.3. IAN Increases the Viability of Tumour Cells

To investigate if IAN influences cell viability, MCF-7 and A375 cells were exposed to 1, 15 and 30 µM of IAN, in the absence of carbidopa. Surprisingly, IAN increased the viability of A375 cells in a concentration-dependent manner (Figure 7A), while in MCF-7 cells, an increase in viability was only observed in response to 15 µM IAN (Figure 7B).

## 4. Discussion

The study of cell metabolism is useful to understand the mechanisms of tumour formation, if they vary along tumour development and treatments, and to better understand tumorigenesis and resistance to treatments. Carbidopa is a drug used combined with L-DOPA to treat PD. It may have anticancer effects. This possibility was raised by the observations that PD patients taking carbidopa have a lower incidence of several cancers [7]. Carbidopa was shown to reduce the viability of BxPC-3 and Capan-2 cells [5]. Based on this observation and because carbidopa is a well-tolerated drug with a long clinical use, carbidopa was proposed for the treatment of pancreatic cancer [5]. Identical arguments have been presented to propose carbidopa for the treatment of prostate cancer [11,12]. 

Breast cancer and melanoma are among the forms of cancer whose incidence may increase in PD patients taking medications containing carbidopa [7]. In the present study, we observed that carbidopa did not reduce the viability of A375 cells and even increased the viability of breast cancer MCF-7 cells (Figure 5), denoting a distinct pattern of response than that described for pancreatic and prostate cell lines. It has to be noticed that the contact of cells with the drug (only 24 h) cannot be compared with the chronic exposure to it in PD treated patients. In the present study, exposure of breast cancer MCF-7 and melanoma A375 cells to carbidopa caused a modification in Trp metabolism, with the appearance of a new Trp metabolite not expected to be produced by mammalian cells: IAN. The pathway involved in IAN synthesis is fully operating in plants and bacteria [10]. The present study shows that the same pathway may be activated in human cancer cell lines when exposed to carbidopa. This illustrates the capacity of cancer cells to mobilize pathways that are usually found in very different branches of the phylogenetic tree.

In plants and bacteria, IAN is considered a messenger involved in defence pathways or a precursor of phytohormones [10,13]. IAN also seems to be produced in breast cancer MCF-7 and melanoma A375 cells when exposed to a drug (carbidopa) commonly used in the chronic treatment of patients with PD, which makes IAN a potential contributor to increased cell viability and, thus, tumorigenesis, in breast and skin. 

## 5. Conclusions

Here, we suggest a possible pathway of Trp metabolism in two cell lines, the IAN pathway, revealed by the inhibition of other alternative pathways by carbidopa (Figure 8). Whether this explains the different incidence of breast cancer and melanoma, compared with other tumours, in PD patients taking carbidopa remains to be confirmed. However, this observation raises some awareness and indicates the need to closely follow signs of breast cancer and melanoma in patients administered carbidopa as a repurposed drug.

## Figures and Tables

**Figure 1 biomolecules-09-00409-f001:**
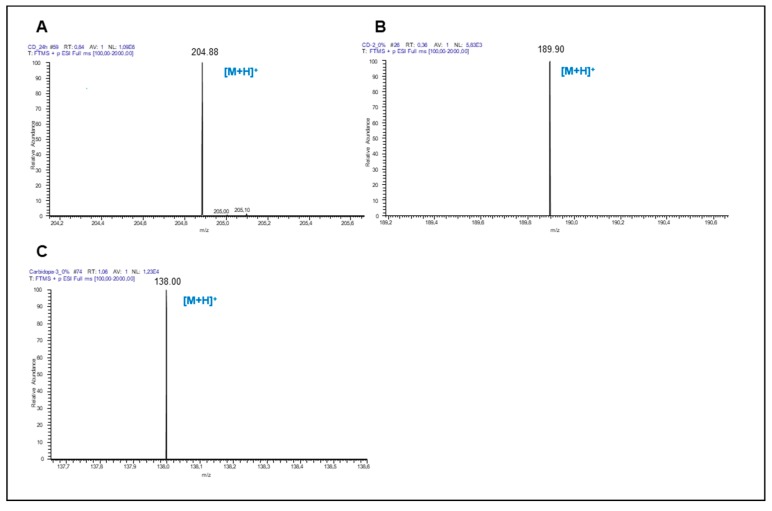
LC-MS detection of tryptophan (Trp) metabolites produced by A375 cells. Mass spectrum (LC-MS, positive mode) of (**A**) Trp (*m/z* 204.09), (**B**) kynurenic acid (*m/z* 189.04) and (**C**) anthranilic acid (*m/z* 137.05) in A375 cells.

**Figure 2 biomolecules-09-00409-f002:**
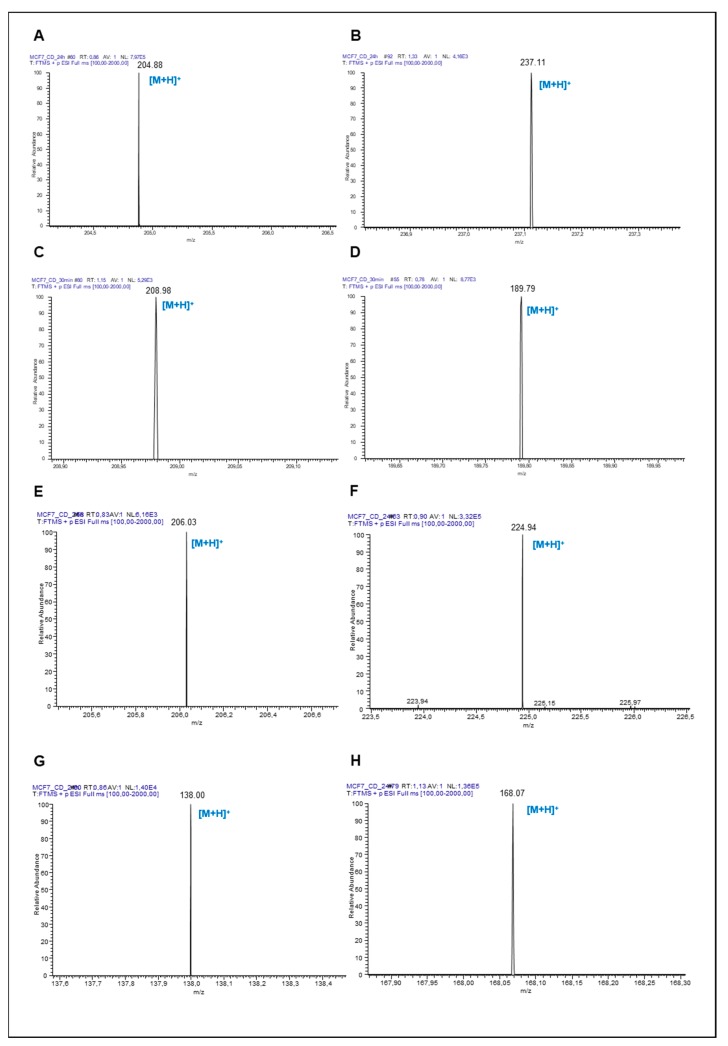
LC-MS detection of Trp metabolites produced by MCF-7 cells. Mass spectrum (LC-MS, positive mode) of (**A**) Trp (*m/z* 204.09), (**B**) *N*-formylkynurenine (*m/z* 236.08), (**C**) kynurenine (*m/z* 208.08), (**D**) kynurenic acid (*m/z* 189.04), (**E**) xanthurenic acid (*m/z* 205.04), (**F**) 3-hydroxykynurenine (*m/z* 224.08), (**G**) anthranilic acid (*m/z* 137.05) and (**H**) quinolinic acid (*m/z* 167.02) in MCF-7 cells.

**Figure 3 biomolecules-09-00409-f003:**
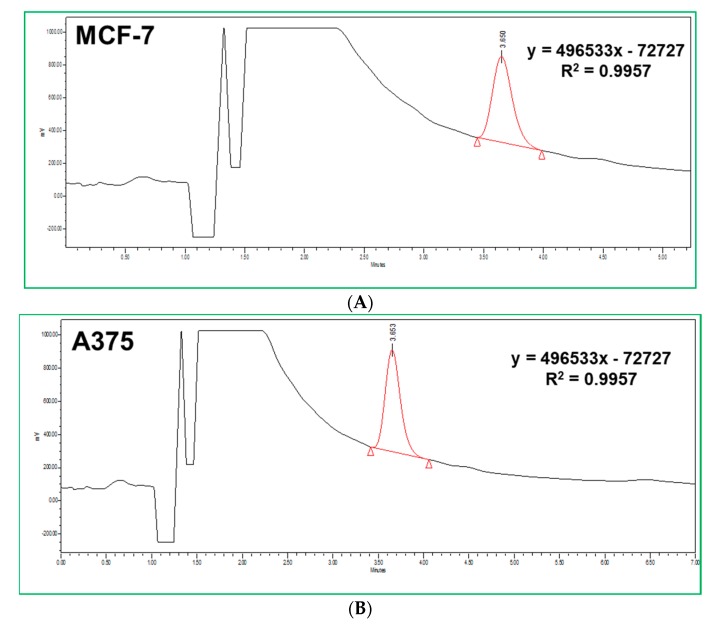
HPLC-ECD quantification of serotonin produced by MCF-7 and A375 cells. (**A**) Equation of the HPLC calibration curve of serotonin and its determination in MCF-7 cells. A concentration of 12.4 nM was detected at 3.650 min, with a peak area of 6,081,666. (**B**) Equation of the HPLC calibration curve of serotonin and its determination in A375 cells. A concentration of 14.6 nM was detected at 3.653 min, with a peak area of 7,155,047. The chromatographic determination was determined using an isocratic HPLC system with electrochemical detection. Calibrator and quality controls were available separately.

**Figure 4 biomolecules-09-00409-f004:**
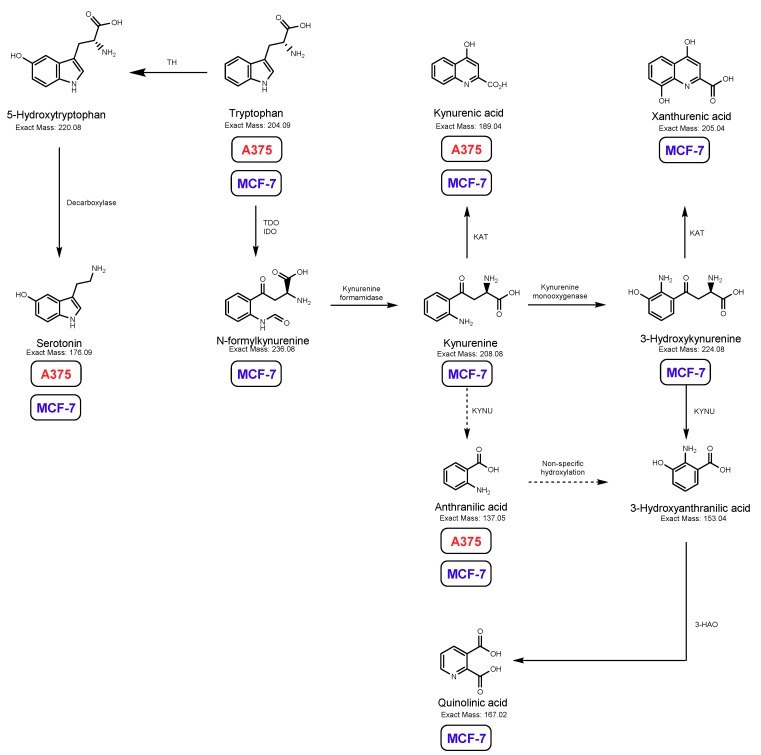
Main metabolites of Trp and their detection in the two cell lines studied, MCF-7 (blue) and A375 (red). Metabolite detection was done by LC-MS, except for serotonin which was detected by HPLC. MCF-7 breast cancer cells metabolize more extensively Trp than A375 melanoma cells, as shown by the larger series of Trp metabolites identified in the medium of MCF-7 cells. TDO: tryptophan 2,3-dioxygenase; IDO: indoleamine 2,3-dioxygenase; KAT: kynurenine aminotransferase; KYNU: kynureninase; 3-HAO: 3-hydroxyanthranilate oxidase.

**Figure 5 biomolecules-09-00409-f005:**
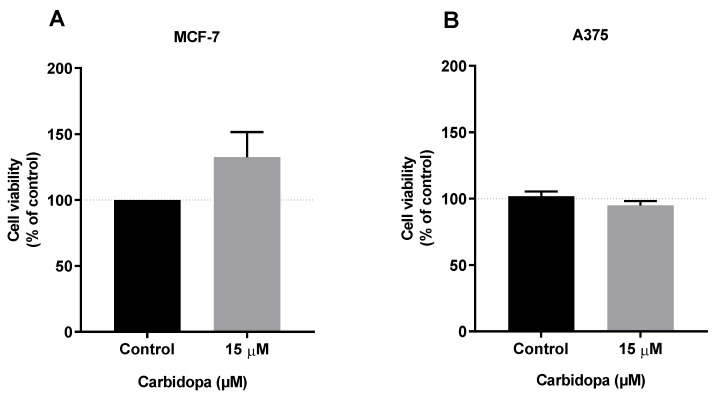
Carbidopa is not cytotoxic in breast and melanoma cell lines. Effect of carbidopa (15 µM) on the viability of (**A**) MCF-7 and (**B**) A375 cells. The cells were treated with carbidopa for 24 h, and then cell viability was determined using the MTT assay. Results are expressed as percentage of the vehicle-treated control ± SEM of six separate experiments.

**Figure 6 biomolecules-09-00409-f006:**
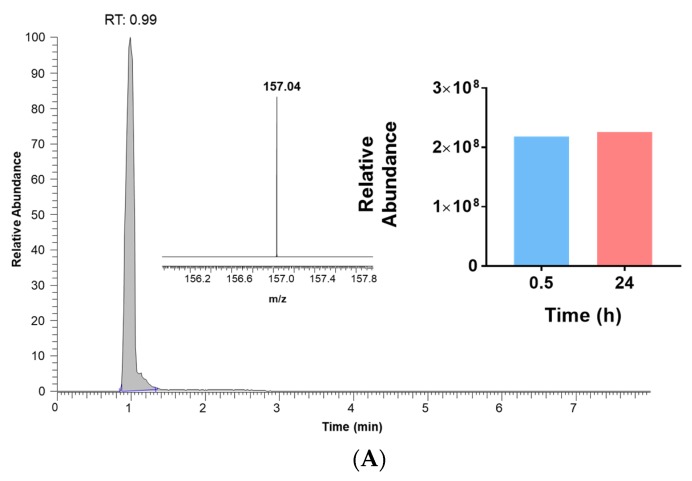
The relative abundance of indole-3-acetonitrile (IAN) is higher in MCF-7 than in A375 cells. LC-MS chromatogram and MS spectrum of the extracellular media of (**A**) A375 and (**B**) MCF-7 cells after (t = 24 h) treatment with carbidopa and relative abundance of IAN (*m/z* 157.04) over 24 h. The grey area indicates the retention time of IAN (*m/z* 157.04, as positive ion mode M + H).

**Figure 7 biomolecules-09-00409-f007:**
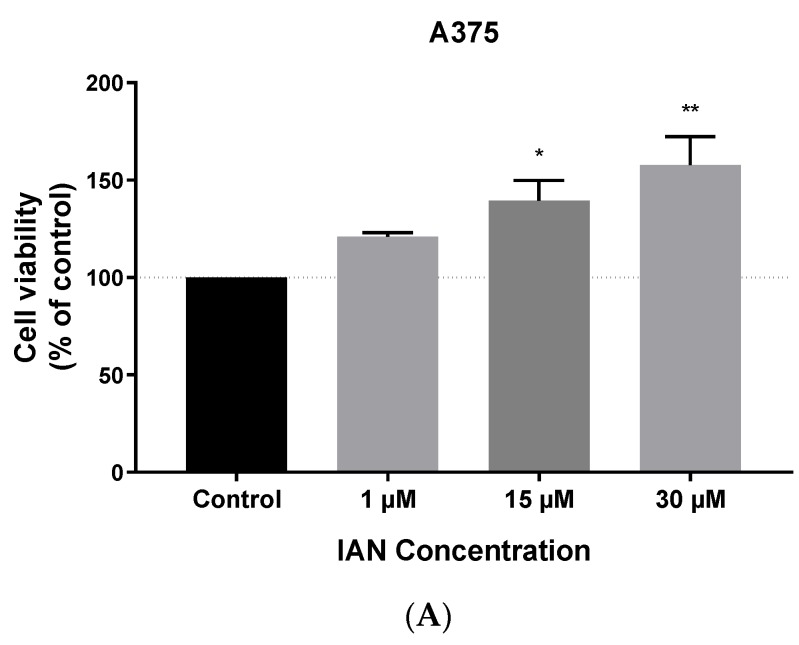
IAN promotes cell survival in a melanoma cell line and has no effect in a breast cancer cell line. Viability assays of (**A**) A375 and (**B**) MCF-7 cells exposed to increasing concentrations of IAN for 24 h. Results are presented as arithmetic means ± SEM (N = 6). (* *p* < 0.05 and ** *p* < 0.01).

**Figure 8 biomolecules-09-00409-f008:**
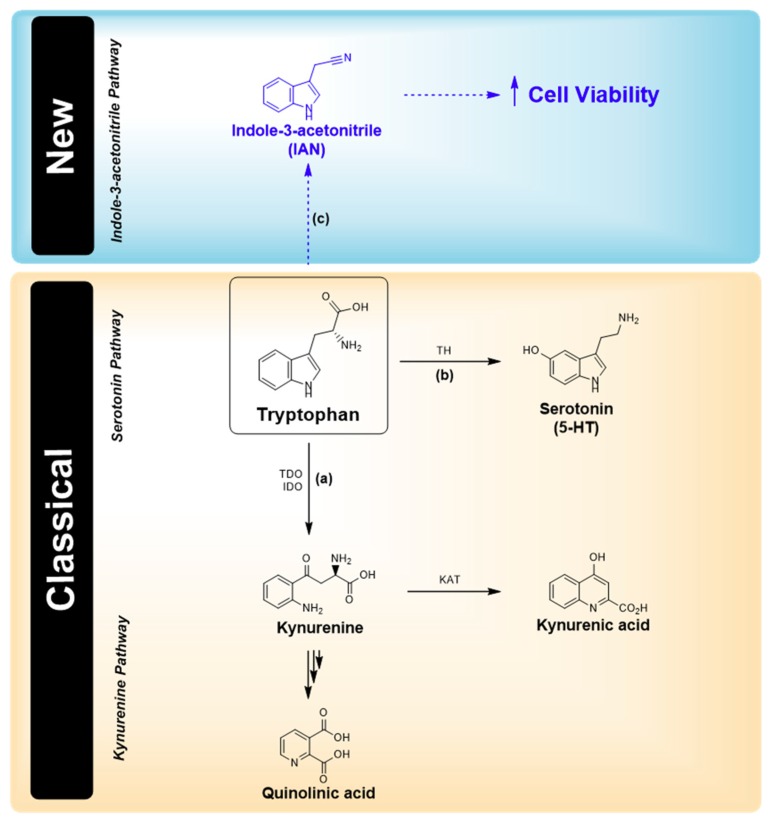
Main metabolic pathway (classical) for Trp in cancer cells and new route proposed in this project. Trp is mostly metabolized by the kynurenine pathway (KP), originating quinolinic acid and kynurenic acid as the main metabolites (a). Trp is also the main precursor of serotonin (5-HT) (b). In this study, we propose the existence of a new alternative route (c) of Trp metabolism leading to the production of IAN.

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
