# Peer review of "Carbidopa Alters Tryptophan Metabolism in Breast Cancer and Melanoma Cells Leading to the Formation of Indole-3-Acetonitrile, a Pro-Proliferative Metabolite"

_biomolecules, 2019, doi:10.3390/biom9090409_

Round 1
Reviewer 1 Report
The article "Carbidopa alter tryptophan metabolism in breast cancer and melanoma cells leading to the formation of indole-3-acetonitrile, a pro-proliferative metabolite" by Duarte et al describes measurements of tryptophan metabolism intermediates in a melanoma and a breast cancer cell line, with and without the addition of carbidopa. The study is based on the observation that PD patients treated with carbidopa show differential tendency to develop breast and skin cancer. The study is in principle quite interesting. There are a few issues or suggestions from my point of view:
Figure 1 and 2 show detection of tryptophan metabolite intermediates in the untreated cells. Could we have that data in a quantified form (LCMS peak areas normalized by cell count/cell volume/protein content or similar, from technical and biological repeats)? In the present form I can only derive that in one measurement there were peaks present indicating the metabolites. That seems to me relatively weak.
How were samples for LCMS prepared, in particular, did the authors use any standards to make sure that the peaks are what they should be? In particular, did they use a standard for IAN?
In figure 6, again I cannot see data from repeat experiments, and I cannot discern what type of normalization, if any, was used. Since the authors claim to find this metabolite for the first time in human cells, I would prefer to see this described in more detail.
I am no expert on tryptophan metabolism: is there a specific reason to measure only extracellular metabolites, or is this for experimental reason?
In figure 7, "promotes cell survival in breast and melanoma cell lines" should be corrected since I cannot discern any signifcant effect on viability in the breast cancer cells.
If carbidopa is metabolized to IAN, and IAN improves viability of the melanoma cells in dose-dependent manner, then carbidopa might also improve viability of the cells in dose-dependent manner; only one dose of carbidopa was tested though. It would be nice to see a carbidopa dose dependent measurement of IAN and viability.
As always, a couple more cell lines would be a good addition esp since breast cancer comes with many subtypes. It might also be an idea to include a cell type of one of the other cancers that seem to be suppressed by carbidopa treatment, as a control. However, I consider this a minor issue, since the initial discovery of the pathway would be quite novel in the present cell types already.
Author Response
Reviewer #1:
The article "Carbidopa alter tryptophan metabolism in breast cancer and melanoma cells leading to the formation of indole-3-acetonitrile, a pro-proliferative metabolite" by Duarte et al describes measurements of tryptophan metabolism intermediates in a melanoma and a breast cancer cell line, with and without the addition of carbidopa. The study is based on the observation that PD patients treated with carbidopa show differential tendency to develop breast and skin cancer. The study is in principle quite interesting. There are a few issues or suggestions from my point of view:
We thank the reviewer for their frankly positive opinion on our work, as well as for their careful revision, comments, and suggestions, which helped us to significantly improve our manuscript. The latter has been revised to meet the reviewers’ comments and suggestions, as follows.
Figure 1 and 2 show detection of tryptophan metabolite intermediates in the untreated cells. Could we have that data in a quantified form (LCMS peak areas normalized by cell count/cell volume/protein content or similar, from technical and biological repeats)? In the present form I can only derive that in one measurement there were peaks present indicating the metabolites. That seems to me relatively weak.
The LC-MS/MS methods he uses provide high enough selectivity and sensitivity to analyze the parent drugs together with its metabolites (others papers from our Research Group). During LC-MS/MS operation, a compound will be ionized and fragmented to specific product ions of the compound. In addition, LC-MS/MS has a very high level of sensitivity to detect trace amounts of the compound. Detection and quantitation was based on the mass-to-charge ratios (m/z) of precursor-product ion pairs, which are specific to compound its metabolites. The number of cells has been standardized and the relative abundance of the compounds is only relative to what is in each sample with a value equal to or greater than 10%. In this way, we can ensure that the peaks studied have sufficient relative abundance to be detected with stability during the analysis.
How were samples for LCMS prepared, in particular, did the authors use any standards to make sure that the peaks are what they should be? In particular, did they use a standard for IAN?
Thank you for the observation. We used the High Resolution Mass Spectral Database [European MassBank (NORMAN MassBank): http://massbank.eu/MassBank/) to compare with standarts and for IAN, we tested the standard (also used in experiments with cells – Figure 7) and the results were similar: exact mass of 156.04.
For monoisotopic exact masses of molecular ion (IAN), in Figure 6, we complete the IAN value information because what is observed is the positive ion mode of [M+H], or [M+1.007276].
Now, we can read in legend of Figure 6:
“Figure 6. The relative abundance of IAN was higher in MCF-7 than in A375 cells. LC-MS chromatogram and MS spectrum of extracellular medium of (A) A375 and (B) MCF-7 cells after (t=24h) contact with carbidopa and relative abundance of IAN (m/z 157.04) over 24 hours. The grey area indicates the retention time IAN (m/z 157.04, as positive ion mode M+H).”
In figure 6, again I cannot see data from repeat experiments, and I cannot discern what type of normalization, if any, was used. Since the authors claim to find this metabolite for the first time in human cells, I would prefer to see this described in more detail.
Yes, we understand. The interpretation for this was developed in the first question about this revision. We used 3/4 samples for experiments in LC-MS and the results was the same between them.
I am no expert on tryptophan metabolism: is there a specific reason to measure only extracellular metabolites, or is this for experimental reason?
The analysis of intracellular metabolites is particularly challenging in cells. Environmental perturbations may considerably affect metabolism/stability of these molecules, which results in intracellular metabolites being rapidly degraded or metabolized by enzymatic reactions. Therefore, quenching or the complete stop of cell metabolism is a pre-requisite for accurate intracellular metabolite analysis. After quenching, metabolites need to be extracted from the intracellular compartment. The choice of the most suitable metabolite extraction method/s is another crucial step. The literature indicates that specific classes of metabolites are better extracted by different extraction protocols. In this sense, the analysis of the extracellular environment is much safer to identify known or new metabolists.
In figure 7, "promotes cell survival in breast and melanoma cell lines" should be corrected since I cannot discern any signifcant effect on viability in the breast cancer cells.
Yes, we agree. Now, we can read in legend of Figure 7:
“Figure 7. IAN promotes cell survival in melanoma cell lines and indifference effect in breast cell lines. Viability assays of (A) A375 and (B) MCF-7 cells exposed to increasing concentrations of IAN for 24 h. Results are presented as arithmetic means ± SEM (N=6). (*P < 0.05 and **P < 0.01).”
If carbidopa is metabolized to IAN, and IAN improves viability of the melanoma cells in dose-dependent manner, then carbidopa might also improve viability of the cells in dose-dependent manner; only one dose of carbidopa was tested though. It would be nice to see a carbidopa dose dependent measurement of IAN and viability.
The carbidopa in not metabolized to IAN, only Trp (see Figure 8).
From literature (Clinical Cancer Research, 2000, 6, 4368-4372) we observed the effect of increasing concentrations of carbidopa on human lung carcinoid (NCI-H727) activity, and the IC50 is ~ 15 microM. Plese, see the Figure. (from E/Figure 3).
Also, the IC50 of carbidopa in aromatic-L-amino acid decarboxylase (AAAD) (in human Carcinoid cells) was 16 microM (Table 1, Clinical Cancer Research, 2000, 6, 4368-4372).
As always, a couple more cell lines would be a good addition esp since breast cancer comes with many subtypes. It might also be an idea to include a cell type of one of the other cancers that seem to be suppressed by carbidopa treatment, as a control. However, I consider this a minor issue, since the initial discovery of the pathway would be quite novel in the present cell types already.
We really appreciate this question. Thank you. From literature, we know that carbidopa is and agonist for AhR. The dose–response data show that Carbidopa activates AhR (aryl hydrocarbon receptor) at concentrations of 3-10 μM, and these concentrations are therapeutically relevant (Biochemical Journal 2017, 474, 3391-3402). Currently, the prevailing notion is that AhR and its ability to induce selective cytochrome P450 enzymes are primarily related to xenobiotic detoxification and to the efficacy of drugs, including anticancer drugs (BMC Cancer 2009, 9, 187). But, recent studies have uncovered a critical role for AhR in cancer (Biochim. Biophys. Acta, Rev. Cancer 2013, 1836, 197-210) and that AhR activation is effective to treat many types of cancers. This type of cells can be evaluated in the future to compare the formation of IAN and the association with aryl hydrocarbon receptor.
More recently, we tested IAN to PC-3 cancer cells and we observed some effect between 1-30 microM of carbidopa. We concluded that the IAN promotes survival in breast and melanoma cancer cells but decrease viability in PC-3 cancer cell lines (see Figure: effect of IAN concentration in viability of PC-3 cell lines).

Reviewer 2 Report
The authors report that carbidopa, a drug for Parkinson disease alters tryptophan metabolism and increases IAN (indole-3-acetonitrile) in MCF-7 breast cancer cells and A375 melanoma cells. The authors present results that IAN increases cell viability of A375 cells and carbidopa does that of MCF-7 cells. Based on the results, they propose that pro-proliferative effect of carbidopa should be ascribed to IAN produced in cells treated with carbidopa. Distinct tryptophan metabolism in breast cancer cells and melanoma cells upon carbidopa treatment is suggested to be a mechanism that might explain peculiar effect of the Parkinson disease drug on breast cancer and melanoma.
Mechanism behind specific pro-proliferative effect of carbidopa on breast cancer and melanoma should be an interesting subject to draw attention of medical society. The authors appears to analyze tryptophan metabolites from cells treated with carbidopa successfully. However, the effect on cell viability of carbidopa and IAN does not match enough to support the conclusion. The effect of carbidopa and IAN is different from each other depending on cell types, which undermines IAN association with carbidopa treatment. Further support for the conclusion should be obtained by comparison of tryptophan metabolite production of breast cancer cells and melanoma cells with other cancer cells whose proliferation are inhibited by carbidopa.
Author Response
Reviewer #2:
The authors report that carbidopa, a drug for Parkinson disease alters tryptophan metabolism and increases IAN (indole-3-acetonitrile) in MCF-7 breast cancer cells and A375 melanoma cells. The authors present results that IAN increases cell viability of A375 cells and carbidopa does that of MCF-7 cells. Based on the results, they propose that pro-proliferative effect of carbidopa should be ascribed to IAN produced in cells treated with carbidopa. Distinct tryptophan metabolism in breast cancer cells and melanoma cells upon carbidopa treatment is suggested to be a mechanism that might explain peculiar effect of the Parkinson disease drug on breast cancer and melanoma.
We thank the reviewer for the observations; we also acknowledge the time spent in the review and the perceptive comments that will undoubtedly improve the quality of the manuscript. The response is addressed below.
Mechanism behind specific pro-proliferative effect of carbidopa on breast cancer and melanoma should be an interesting subject to draw attention of medical society. The authors appears to analyze tryptophan metabolites from cells treated with carbidopa successfully. However, the effect on cell viability of carbidopa and IAN does not match enough to support the conclusion. The effect of carbidopa and IAN is different from each other depending on cell types, which undermines IAN association with carbidopa treatment. Further support for the conclusion should be obtained by comparison of tryptophan metabolite production of breast cancer cells and melanoma cells with other cancer cells whose proliferation are inhibited by carbidopa.
We really appreciate this question. Thank you. From literature, we know that carbidopa is and agonist for AhR. The dose–response data show that Carbidopa activates AhR (aryl hydrocarbon receptor) at concentrations of 3-10 μM, and these concentrations are therapeutically relevant (Biochemical Journal 2017, 474, 3391-3402). Currently, the prevailing notion is that AhR and its ability to induce selective cytochrome P450 enzymes are primarily related to xenobiotic detoxification and to the efficacy of drugs, including anticancer drugs (BMC Cancer 2009, 9, 187). But, recent studies have uncovered a critical role for AhR in cancer (Biochim. Biophys. Acta, Rev. Cancer 2013, 1836, 197-210) and that AhR activation is effective to treat many types of cancers. This type of cells can be evaluated in the future to compare the formation of IAN and the association with aryl hydrocarbon receptor.
More recently, we tested IAN to PC-3 cancer cells and we observed some effect between 1-30 microM of carbidopa. We concluded that the IAN promotes survival in breast and melanoma cancer cells but decrease viability in PC-3 cancer cell lines (see Figure: effect of IAN concentration in viability of PC-3 cell lines).

Round 2
Reviewer 2 Report
The revision includes changes in the legend of Fig 7 which now appears corresponding to the results shown. The authors also present results on the effect or IAN in PC-3 cells as a reference. The key of current report is understood to be that carbidopa alters trp metabolism in breast cancer and melanoma cells to produce IAN, which, in turn, might enhance cell viability of the cells. Although the authors did attempt to address the reviewer’s concern by rephrasing the figure legend and showing PC-3 result, there are still some paucity to support their claim and to warrant publication as it is.
There is a discrepancy in the effect of carbidopa and IAN on the cell viability of breast cancer and melanoma cells, respectively, which was pointed out in the previous review (Fig 5 vs. Fig. 7). It is not clear whether the effect of carbidopa on cell viability is mediated by IAN. The authors are required to explain the difference and their implication in the effect of carbidopa on these cells. They show viability results at 24 h, which could be too short to develop enough effect of carbidopa or IAN. Extending time period of treatment might be an option.
In conjunction with the above notion, IAN did not seem to increase in A375 cells in Fig 6. Line 211 of page 8 says that IAN was increased at early time point. In conjunction with the notion, the authors are required to provide results showing increase of IAN upon carbidopa treatment such as changes of IAN with time after treatment of carbidopa.
It is still not clear whether carbidopa alters trp metabolism specifically in A375 and MCF-7 cells to produce IAN or whether IAN by itself has distinct effect on viability of these cells including PC-3 cells. It is recommended to analyze the effect of carbidopa on the viability of PC-3 and IAN production in PC-3 cells in parallel with A375 and MCF-7 cells.
Author Response
Reviewers' comments:
Reviewer #2:
The revision includes changes in the legend of Fig 7 which now appears corresponding to the results shown. The authors also present results on the effect or IAN in PC-3 cells as a reference. The key of current report is understood to be that carbidopa alters trp metabolism in breast cancer and melanoma cells to produce IAN, which, in turn, might enhance cell viability of the cells. Although the authors did attempt to address the reviewer’s concern by rephrasing the figure legend and showing PC-3 result, there are still some paucity to support their claim and to warrant publication as it is.
We thank the reviewer for the observations; we also acknowledge the time spent in the review and the perceptive comments that will undoubtedly improve the quality of the manuscript. The response is addressed below.
There is a discrepancy in the effect of carbidopa and IAN on the cell viability of breast cancer and melanoma cells, respectively, which was pointed out in the previous review (Fig 5 vs. Fig. 7). It is not clear whether the effect of carbidopa on cell viability is mediated by IAN. The authors are required to explain the difference and their implication in the effect of carbidopa on these cells. They show viability results at 24 h, which could be too short to develop enough effect of carbidopa or IAN. Extending time period of treatment might be an option.
We really appreciate this question. We prefer to have a 24-hour study to maintain a link to clinical carbidopa use. In addition, previous studies in our Group suggest that the main effects occur around 24 hours.
The difference in the effect of carbidopa on these cells is probably associated to mechanism of action. Carbidopa is an inhibitor of the amino acid decarboxylase (DDC) which in order, inhibits the peripheral metabolism of levodopa. DDC is very important in the biosynthesis of L-tryptophan to serotonin and the modification of L-DOPA to dopamine. DDC can be found in the body periphery and in the blood-brain barrier.
The LC-MS/MS methods he uses provide high enough selectivity and sensitivity to analyze the parent drugs together with its metabolites (others papers from our Research Group). During LC-MS/MS operation, a compound will be ionized and fragmented to specific product ions of the compound. In addition, LC-MS/MS has a very high level of sensitivity to detect trace amounts of the compound. Detection and quantitation was based on the mass-to-charge ratios (m/z) of precursor-product ion pairs, which are specific to compound its metabolites. The number of cells has been standardized and the relative abundance of the compounds is only relative to what is in each sample with a value equal to or greater than 10%. In this way, we can ensure that the peaks studied have sufficient relative abundance to be detected with stability during the analysis.
In conjunction with the above notion, IAN did not seem to increase in A375 cells in Fig 6. Line 211 of page 8 says that IAN was increased at early time point. In conjunction with the notion, the authors are required to provide results showing increase of IAN upon carbidopa treatment such as changes of IAN with time after treatment of carbidopa.
Thank for this observation. The analysis of intracellular metabolites is particularly challenging in cells. Environmental perturbations may considerably affect metabolism/stability of these molecules, which results in intracellular metabolites being rapidly degraded or metabolized by enzymatic reactions. Therefore, quenching or the complete stop of cell metabolism is a pre-requisite for accurate intracellular metabolite analysis. After quenching, metabolites need to be extracted from the intracellular compartment. The choice of the most suitable metabolite extraction method/s is another crucial step. The literature indicates that specific classes of metabolites are better extracted by different extraction protocols. In this sense, the analysis of the extracellular environment is much safer to identify known or new metabolists.
It is still not clear whether carbidopa alters trp metabolism specifically in A375 and MCF-7 cells to produce IAN or whether IAN by itself has distinct effect on viability of these cells including PC-3 cells. It is recommended to analyze the effect of carbidopa on the viability of PC-3 and IAN production in PC-3 cells in parallel with A375 and MCF-7 cells.
Carbidopa is an inhibitor of the amino acid decarboxylase (DDC) which in order, inhibits the peripheral metabolism of levodopa. DDC is very important in the biosynthesis of L-tryptophan to serotonin and the modification of L-DOPA to dopamine. As we mentioned before, the dose-response data show that carbidopa activates AhR (aryl hydrocarbon receptor) at concentrations of 3-10 μM, and these concentrations are therapeutically relevant (Biochemical Journal 2017, 474, 3391-3402). Currently, the prevailing notion is that AhR and its ability to induce selective cytochrome P450 enzymes are primarily related to xenobiotic detoxification and to the efficacy of drugs, including anticancer drugs (BMC Cancer 2009, 9, 187). But, recent studies have uncovered a critical role for AhR in cancer (Biochim. Biophys. Acta, Rev. Cancer 2013, 1836, 197-210) and that AhR activation is effective to treat many types of cancers. This type of cells can be evaluated in the future to compare the formation of IAN and the association with aryl hydrocarbon receptor. Thank for the observation and suggestion.